# Labelizer: systematic selection of protein residues for covalent fluorophore labeling

Christian Gebhardt [1] ✉, Pascal Bawidamann[1], Anna-Katharina Spring[1,2], Robin Schenk [3], Konstantin Schütze [1], Gabriel G. Moya Muñoz [1,2], Nicolas D. Wendler [1,2], Douglas A. Griffith[1], Jan Lipfert [4,5] ✉ & Thorben Cordes [1,2] ✉

Attaching fluorescent dyes to biomolecules is essential for assays in biology, biochemistry, biophysics, biomedicine and imaging. A systematic approach for the selection of suitable labeling sites in macromolecules, particularly proteins, is missing. We present a quantitative strategy to identify such protein residues using a naïve Bayes classifier. Analysis of >100 proteins with ~400 successfully labeled residues allows to identify four parameters, which can rank residues via a single metric (the label score). The approach is tested and benchmarked by inspection of literature data and experiments on the expression level, degree of labelling, and success in FRET assays of different bacterial substrate binding proteins. With the paper, we provide a python package and webserver (https://labelizer.org), that performs an analysis of a pdb-structure (or model), label score calculation, and FRET assay scoring. The approach can facilitate to build up a central open-access database to continuously refine the label-site selection in proteins.

Microscopy and spectroscopy techniques are ubiquitously used in the life sciences, in biophysical and medical assays to investigate the structure, interactions, and dynamics of macromolecules and their complexes down to the single-molecule level[1–5]. Many applications require specific labeling of the biomolecule of interest with fluorescent probes[6–12]. Whereas fluorescent proteins are the first choice for imaging applications in live-cells[13–15], synthetic organic fluorophores (dyes) are often used for high-sensitivity applications including single-molecule detection[16–18] and super-resolution microscopy[19–21]. A common strategy for the (covalent) attachment of functional probes to proteins, including dyes, EPR spin probes, nanoparticles, and reactive surfaces is via reactive linker moieties[6,22].

A range of labeling strategies exists that exploit reactive groups, each with unique (dis)advantages. Coupling to amino groups in lysine residues can be achieved via N-hydroxysuccinimide (NHS)-esters, but this approach lacks specificity because of the abundance of lysine residues in proteins[22]. Alternatively, a terminally located His-tag or the N-terminus of the protein itself can be used for selective attachment of functional probes, with the disadvantage that the choice of labeling position is greatly curtailed[22]. In contrast, peptide tags (e.g., CLIP, SNAP, Halo, etc.) can facilitate covalent or enzymatic probe attachment (AP-BirA, LPXTG-SortaseA, etc.) at any desired location, but the size of tags limits applications and can impact protein function[23]. The most widely used strategy for site-specific labeling of proteins is, therefore, to introduce non-native cysteine residues and to label their sulfhydryl-moiety via a maleimide-conjugate of the selected probe[22,24]. Cysteine residues can be labeled with minimal effects on protein structure and function. Alternatively, unnatural amino acids (UAAs)

[1]Physical and Synthetic Biology, Faculty of Biology, Ludwig-Maximilians-Universität München, Großhadernerstr. 2-4, Planegg-Martinsried, Germany. [2]Biophysical Chemistry, Department of Chemistry and Chemical Biology, Technische Universität Dortmund, Dortmund, Germany. [3]Klinikum rechts der Isar, Technische Universität München, Klinik und Poliklinik für Innere Medizin II, München, Germany. [4]Department of Physics and Center for NanoScience, Ludwig-Maximilians-Universität München, Amalienstr. 54, München, Germany. [5]Soft Condensed Matter and Biophysics, Department of Physics and Debye Institute for Nanomaterials Science, Utrecht University, Princetonplein 1, Utrecht, The Netherlands. ✉e-mail: gebhardt.christian@gmx.net; j.lipfert@uu.nl; thorben.cordes@tu-dortmund.de

can be introduced as targets for labeling. UAAs have proven particularly useful in cases where the removal of native cysteines is not possible due to their relevance (or abundance) and for live-cell labeling, where too many different proteins with cysteine residues are present[25–30].

The introduction of cysteine residues or UAAs have become the methods of choice for many spectroscopic and microscopic studies of proteins, including the characterization of structural and functional dynamics by single-molecule Förster resonance energy transfer (smFRET)[28,31,32] or pulsed electron-electron double resonance spectroscopy (PELDOR or DEER)[33–36]. Therefore, the ability to select optimal labeling sites for the introduction of suitable probes has grown in importance[37–39]. Currently, labeling sites are typically selected based on manual inspection of the protein structure in a lengthy trial and error process to identify labeling sites via physicochemical intuition that are not essential for protein structure or function[40–49], but that are also compatible with the assay requirements, e.g., for FRET to result in an inter-fluorophore distance close to the Förster Radius $R_0$[28,31]. Frequently encountered problems when selecting a labeling site for fluorescent dyes (Fig. 1A) range from (i) influence of the fluorophore on protein properties, including altered biochemical function (Fig. 1A,

"Protein"), (ii) low labeling efficiency (Fig. 1A, "Labelling efficiency"), or (iii) unwanted dye-protein interactions (Fig. 1A, "Dye Orientation"), to (iv) unpredictable or unfavorable photophysical properties of the dyes at the chosen site (Fig. 1A, "Spectroscopic Properties"). Suitable residues for labeling must not only enable specific and efficient attachment of fluorophores, but also avoid the problems summarized in Fig. 1A. Currently, the selection of labeling sites is often based on sensible rules of thumb[50] selecting those residues that satisfy assay requirements (e.g., distance constraints for FRET[51–54]), but that are also solvent accessible[55], show low conservation scores[28] and are not related to protein function or the presence of fluorescence quenchers such as tryptophans[50,51,53,54,56].

Here, we introduce an automated analysis pipeline based on a naïve Bayes classifier[57,58] to select suitable label sites using information of protein structure and sequence, e.g., from the protein data bank, PDB (Fig. 1C, step 1). To systematically compare sites, we introduce a quantitative *label score LS*, which indicates the suitability of a protein residue to become a label-site, at which any of the problems shown in Fig. 1A are minimal. We assembled a database of publications that report the successful labeling of protein variants used in biophysical assays and identified an ideal set of parameters to allow the ranking of

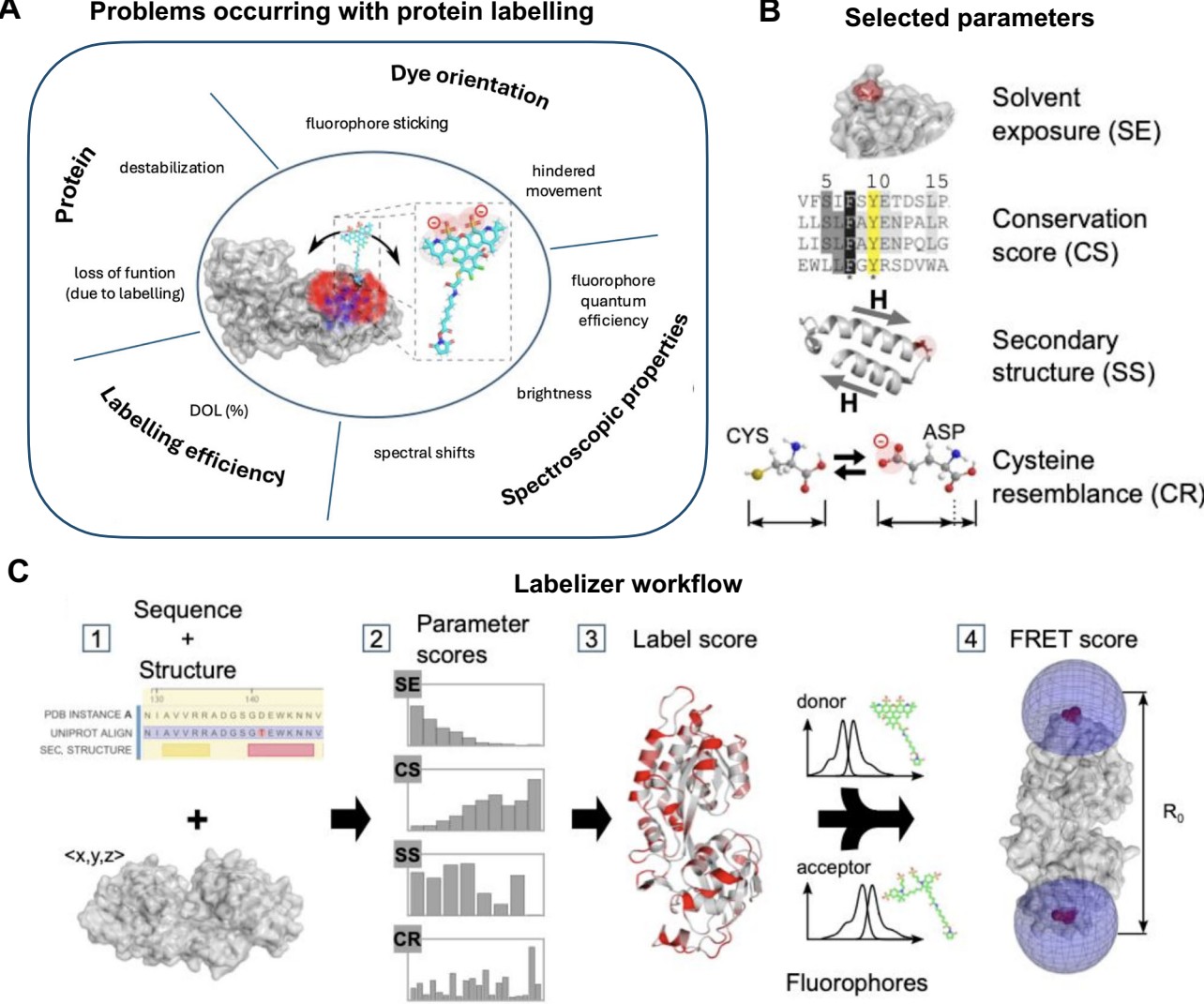

**Fig. 1 | Labelizer workflow to score protein residues for labeling and FRET experiments. A** Schematic overview of protein-fluorophore interactions that can impact the quality and success of fluorescence assays. **B** Parameter categories obtained from protein structures and databases used for the labelizer analysis. **C** Workflow for identifying suitable labeling sites, label score representation and selecting residue pairs for FRET experiments with FRET scoring.

such residues (Fig. 1C, step 2/3). *LS* can be calculated independently of the choice of the label (fluorophore, EPR probe, beads, surfaces, etc.), yet we here focus on the use and characterization of *LS* for the attachment of fluorescent dyes to proteins. We also extended our analysis to pairs of residues for FRET assays, where the interdye distance should be close to the Förster radius to obtain the highest sensitivity (Fig. 1C, step 4). Therefore, we score different residue pairs according to *LS* and simulated distances to obtain an optimal FRET assay, which express the suitability of a residue pair as a FRET score. We support the predictive power of the *LS* and FRET scores with data from the literature and experiments on substrate-binding proteins (SBPs)[59–61].

To make the analysis routine available to a large community of researchers, we introduce a python package called "labelizer", which implements our analysis of protein structures, label score calculation, and FRET assay scoring. The labelizer package allows researchers to build on our findings and adapt the code for their specific needs. For straightforward use, we also provide a webserver (https://labelizer. org) with a user-friendly interface to apply our analysis approach without any programming efforts.

## Results

### Database of successfully labeled residues

As the basis of our label-site selection tool, we created a database of proteins that have been successfully labeled with fluorophores. A large set (>1000) of peer-reviewed papers and preprints was screened for labeled cysteine or UAA residues in proteins. We include protein residues in the database that have been covalently and site-specifically labeled at cysteines (predominantly) or UAAs with organic fluorophores. Note that we also included some spin labels or biotin-linked fluorophores, yet these represent < 5% of all labels in the database (see Supplementary Fig. 1). Furthermore, only residues are included for which the structure of the protein has been deposited in the PDB. For the included proteins, we extract information on the labeled residue (chain, number), the type of mutation used for labeling (cysteine or UAA), the assay type (e.g., single fluorophore assays, smFRET assay with two labels, imaging, bulk FRET, etc.), and the type of label. We then gathered additional information on the protein, such as its oligomeric state (monomer, dimer, complexes), whether the protein structure has been experimentally determined or only a homology model is available, and whether it is a soluble or a membrane protein. Overall, we identified labeled residues in >100 different proteins from >100 publications (see Supplementary Data: Reference Database Labelizer). An overview of the data and summary statistics are presented in Supplementary Fig. 1.

We used a standardized pre-processing routine (see Methods and Supplementary Note 1) to extract all relevant residues from the pdb-files of the proteins in the database. The final data set from 104 pdb structures contains 43357 residues, 396 of which are reported to have been successfully labeled (the other residues are considered unknown). For all residues in our database, we compute multiple parameters that can be assigned to one of the four major categories (Fig. 1B): (i) conservation score CS (ii) solvent exposure SE, (iii) secondary structure SS, and (iv), amino acid similarity of the exchanged residues to a cysteine, which we abbreviate as cysteine resemblance CR (see Supplementary Note 1 with Table 1–6). The parameters are either directly extracted from the residues in question, e.g., amino acid type, mass, charge, and size, or calculated with the help of freely available software (conservation score (ConSurf[62,63]), solvent exposure (DSSP[64], HSE[65], MSMS[66]), and secondary structure (DSSP[64])). Altogether, we obtain 28 parameters for each residue.

### Bayesian approach to the prediction of labeling sites

To identify suitable residues for labeling, we are interested in $P(l|s)$, the conditional probability that the residue can be labeled given a parameter value *s*. By Bayes' law

$$P(l|s) = \frac{P(s|l)}{P(s)} P(l),  \quad (1)$$

$P(s)$ is the probability distribution of the parameter values *s* over all residues, whether or not they can be labeled, while $P(s|l)$ is the probability distribution of the parameter values *s* given that the residue can be labeled. Finally, $P(l)$ is the a priori probability that a residue can be labeled. While $P(s)$ and $P(s|l)$ can be readily computed from our database of labeled protein structures, $P(l)$ is harder to assess since the literature is biased towards reporting successful attempts of labeling that have provided relevant insights. Since $P(l)$ only scales the final probability and does not affect the predictions of the relative ease of labeling for different residues, we decided to here use a simplified parameter score

$$PS(s) = \frac{P(s|l)}{P(s)}  \quad (2)$$

instead of P(l|s) to assess the suitability of residues for labeling. PS(s) is, in essence, the odds ratio for a given parameter value to occur in a labeled residue compared to randomly selected residues. For all 28 parameters, we computed $P(s|l)$ distributions for the 396 successfully labeled residues and $P(s)$ distributions from all 43357 residues of the 112 chains of the database (Fig. 2A and Supplementary Figs. 2, 3).

As a control, we compared the probability distributions $P(s)$ from our database of successfully labeled residues with the distributions computed for a random selection of protein chains from the PDB (PDBselect, November 2017)[67,68] (see "Methods"). Here, we find only minor differences, indicating that the protein parameters in our database are representative of the overall PDB content (Supplementary Fig. 2). One notable difference is that cysteines are much less abundant (by ~ 50%) in the database of labeled proteins compared to the overall PDB, suggesting that cysteine insertion and labeling is easier (or at least more common) for proteins with fewer native cysteines (Supplementary Fig. 2). Although we also included residues that were labeled via UAA incorporation, our database indicates that cysteine labeling is still the predominant strategy for proteins since it makes up ~ 90% of all labeled residues in our database (Supplementary Fig. 1D).

We find clear differences between P(s|l) and P(s) and, therefore, non-uniform PS distributions for most of the investigated parameters (Fig. 2A, C and Supplementary Fig. 6), showing that they indeed contain information about the suitability of residues to serve as label sites. To evaluate which parameters are most predictive, we computed *PS* distributions for 28 parameters (numbered from #1 to #28) from all four categories from our database (Fig. 2 and Supplementary Table 1, 4). For each *PS* distribution, we analyzed their mean-square deviation from an equal distribution, the Gini coefficient, and the Shannon entropy (see Supplementary Note 1 and Supplementary Table 6). We find that the *PS* distributions for many parameters clearly deviate from an equal distribution and contain significant information (low Shannon entropy), e.g., seen in #1: relative surface area (Wilke), #4: first half-sphere exposure (10 Å), #16: variant length in homologs (see Supplementary Fig. 3). Other parameters contain barely any information such as #17 cysteine in homologs (yes/no), or #27 amino acid charge (Supplementary Fig. 3). Thus, strikingly, it is largely unpredictive for labeling of a residue whether a cysteine is found in one of the homolog proteins at the same position or whether the residue is charged (see parameter #17 and #27, Supplementary Fig. 3). One might have expected that residues with cysteine homologs are easily mutated to cysteines, and therefore, significantly enhanced in our scoring, which is not the case.

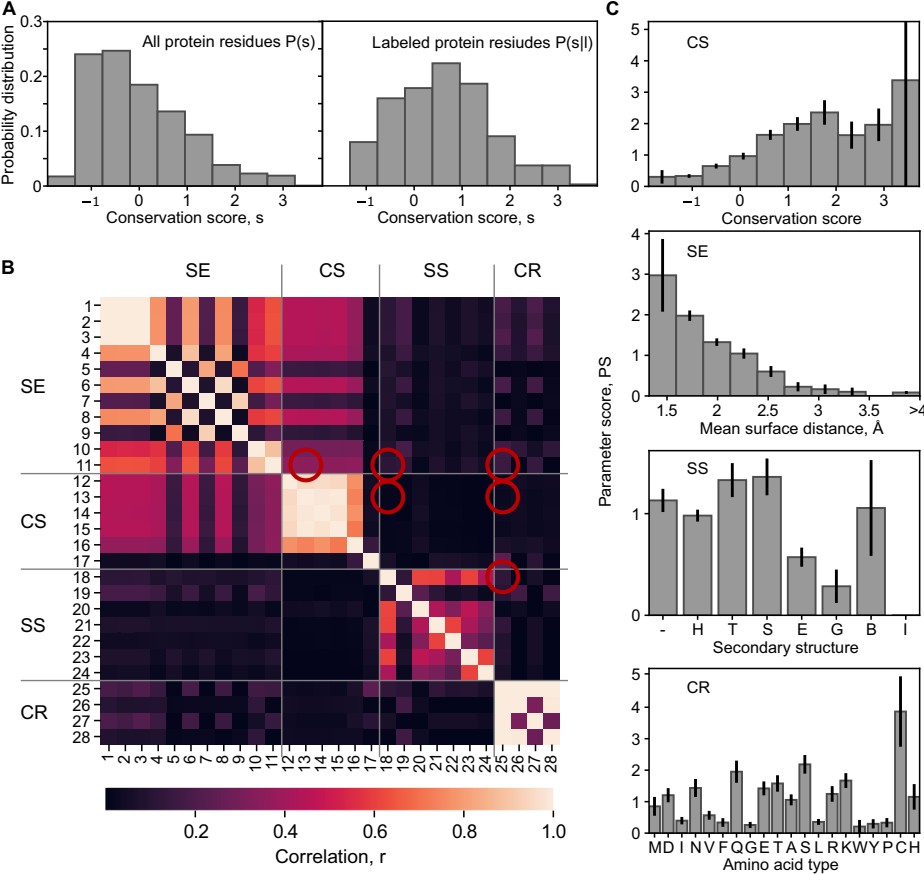

**Fig. 2 | Parameter score analysis. A** Probability distribution $P(s)$ for the parameter ConSurf score (#13, Supplementary Table 2 and 6, negative values represent highly conserved residues among homologs) for all analyzed residues (left) and for the successfully labeled residues $P(s|l)$; right. **B** Correlations between all parameters were calculated based on Pearson correlation (numeric-numeric), interclass correlation (categorical-numeric), or Cramer's V (categorical-categorical). The cross-correlations of the final parameter selection for the labelizer algorithm are marked (red circles). **C** Parameter score distributions $PS = P(s|l)/P(s)$ for the four

parameters that we select as the default for scoring. The top panel shows the resulting parameter score from the distribution in A. For the other categories, the parameters are: solvent exposure (#11, mean surface distance, Supplementary Table 1, 6), secondary structure (#18, secondary structure from DSSP, Supplementary Table 3, 6), and cysteine resemblance (#25, amino acid identity, Supplementary Table 4, 6). Error bars are the standard deviation from counting statistics. The clear deviations from uniform distributions indicate that all four parameters contain information about the suitability of a site for labeling.

After establishing the predictive power of individual parameters, we investigated what combinations of parameters should be used. For this, we calculated the correlation between all parameters to judge their statistical independence, which is desirable for our Bayesian analysis (Fig. 2B). Since we deal with categorical data (e.g. secondary structure) and numerical data (e.g., relative surface area), we used Pearson correlation, interclass correlation and Cramer's V for the combinations of numeric-numeric, categorical-numeric, categorical-categorical values, respectively (see Methods for details). We formed sets of four parameters and used a correlation measure (2-norm of all paired correlations, see Methods) to calculate a combined correlation estimator for all combinations of parameters (Supplementary Fig. 4). Whereas this combined correlation-derived measure shows higher values for most combinations of two or more parameters within the same categories CS, SE, CR, and SS, the correlation of combinations of parameters from different categories was smaller (< 0.5). This effect was independent of whether parameters with high or low predictive power (MSD / Shannon entropy) were combined (Fig. 2B and Supplementary Fig. 4). The overall low correlation between parameters from different categories justifies our categorization and their consideration as independent variables if we restrict our selection to one parameter per category. The strong correlation within categories also suggests that the choice of the particular parameter from one category is not critical, i.e., most of

the parameters can account for the properties of the respective category.

## The combined label score predicts potential labeling sites

To combine parameter scores into a final assessment of a given residue to serve as a label site, we introduce a combined label score, $LS$. By standard probability theory, different parameters $s_i$ can be combined by

$$P\left(l\bigcap_{i=1}^{n}s_i\right) = \prod_{i=1}^{n}P(l|s_i) = \frac{\prod_{i=1}^{n}P(s_i|l)P(l)}{\prod_{i=1}^{n}P(s_i)} \quad (3)$$

under the assumption that they are independent, where $\prod$ denotes the product and $\cap$ the intersection. This naïve Bayes classification[57,58] is known to give good predictions for low and moderately correlated parameters[69–72], which is the case for our parameter set (Fig. 2B). In general, any residual correlation alters the calculated probability values towards the extremes of 0 and 1[72]. However, we again use parameter scores as comparative figures without the meaning of probabilities and combine the $PS_i$ into the combined label score by taking their geometric mean:

$$LS = \sqrt[n]{PS_1 \cdot \ldots \cdot PS_n} \quad (4)$$

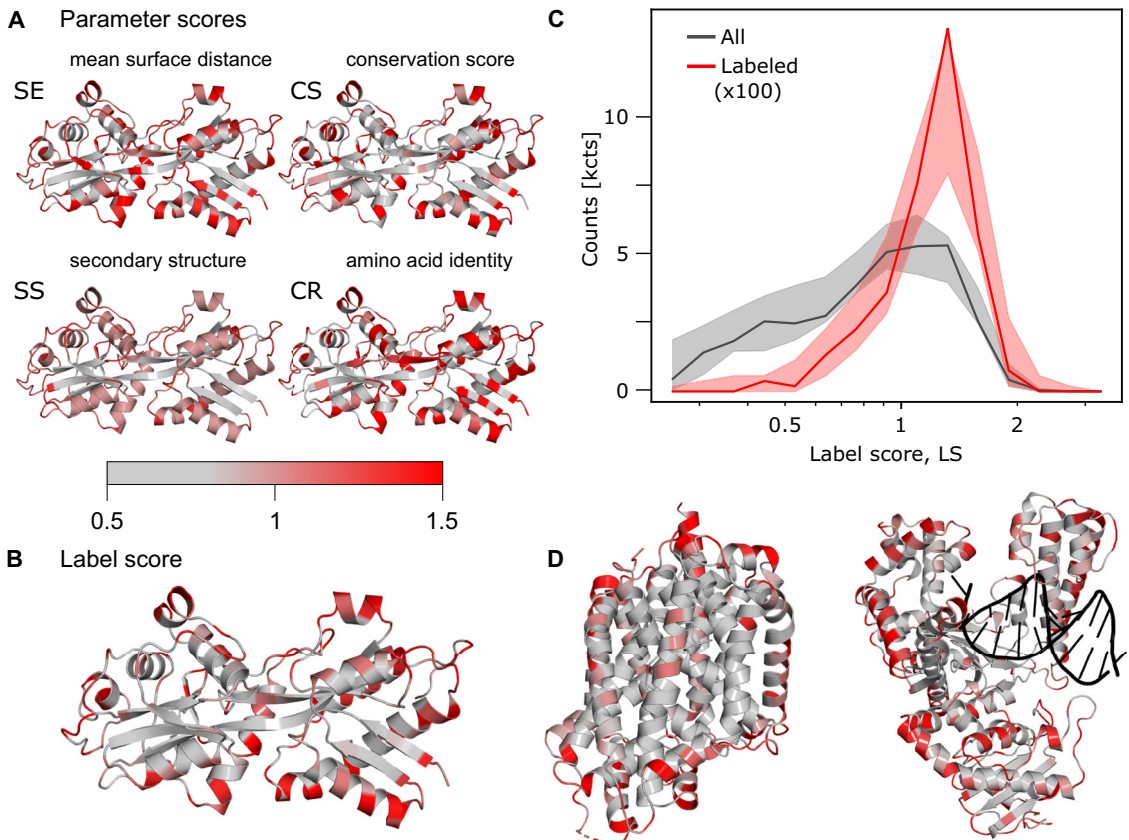

**Fig. 3 | Visualization of parameter and label scores. A** Visualization of the selected parameter scores from the four categories, which are used as default settings in our webserver for the example of PBP from *E.coli*[106,107]; pdb:1OIB. **B** Visualization of the label score on PBP based on default parameters shown in panel (**A**). **C** Label score histogram of all residues (gray) and labeled residues (red) in the database. Due to lower numbers of residues in the labeled data set it was multiplied with a factor of 100 to allow better comparison of the distributions. The shaded area shows the 95% confidence interval from 400 bootstrapping runs. **D** Additional examples of *LS* values indicated on protein structures for a membrane protein (left, LeuT of in A. aeolicus[108,109], pdb:2A65) and a DNA-binding protein (right, DNA polymerase I of B. stearothermophilus[110,111] with DNA template, pdb:1L3U).

An important question is which of the 28 parameters to include in the *LS*. We include one parameter from each of the four categories CS, SE, SR, and SS, for which concrete values were mapped onto the structure of the phosphate binding protein PBP (Fig. 3A). For a rational selection of parameters, we strive (i) to maximize the dynamic range of values for *LS*, (ii) to maximize the enhancement/suppression level of *LS* of the successfully labeled residues in the database for high/low *LS* values and (iii) to maximize the statistical significance level of *LS* values of random residues over *LS* values of the labeled residues in the database.

Based on these criteria, we were able to identify several parameter sets with predictive power (Supplementary Fig. 5A, B), but also combinations with much less information (Supplementary Fig. 5C). In the end, we decided on one set that resulted in a large difference of the distributions between the random and labeled residues: mean surface distance (SE, #11), conservation score (CS, #13), secondary structure of the labeled residue (SS, #18), and the nature of the mutated amino acid (CR, #25). This set is shown in Fig. 3C and is used as the default for *LS* calculations in this manuscript and for the associated web server. In the labelizer Python package, any parameter combination can be selected.

We chose the default set out of all well-performing combinations, because of the intuitive nature of all selected parameters and the maximized differences between the mean *LS* values of all vs. the labeled residues. Both our choice of parameters and the selected number of categories to four (and not only two or three) are supported by statistical analysis of the significance, i.e., a *t* test and a comparison of the mean values of all vs. labeled parameters for different parameter combinations (Supplementary Table 7). Our selection is further validated by comparing the receiver operating characteristic (ROC curve) for the baseline when retraining with one of four scores removed and the predictive power of each of the scores on their own (Supplementary Fig. 6). Bootstrapping of the final set demonstrates the robustness of our analysis (Fig. 3C). For this final set of parameters, we find that the label scores *LS* range from 0.2 to 2 for most residues (except 5% failed calculations with *LS* = 0). The ratio of the *LS* distribution of successfully labeled residues in the database and all label scores shows that high label scores (> 1.5) are significantly enhanced by a factor of ~ 3-4 for the labeled residues, whereas low label scores (< 0.5) are suppressed by a factor of ~ 10 (Fig. 3C). This suggests that the label score is an informative measure to rank and compare residues for their suitability for labeling with fluorophores. We note here that it would be beneficial to compare the label scores of successfully labeled residues with nonsuccessfully labeled residues in the future. However, we do not have information on non-successfully labeled residues, and only 1% of the considered residues (396 out of 43357) are known as labeled, which should not affect the comparison significantly. We visualize the calculated *LS* scores for three typical proteins, comprising a soluble protein, a membrane protein, and a DNA binding protein (Fig. 3D–F).

## Experimental benchmarking of the label score

To characterize the relation between *LS* values and experimentally observed behavior, we performed two different analyses of variants

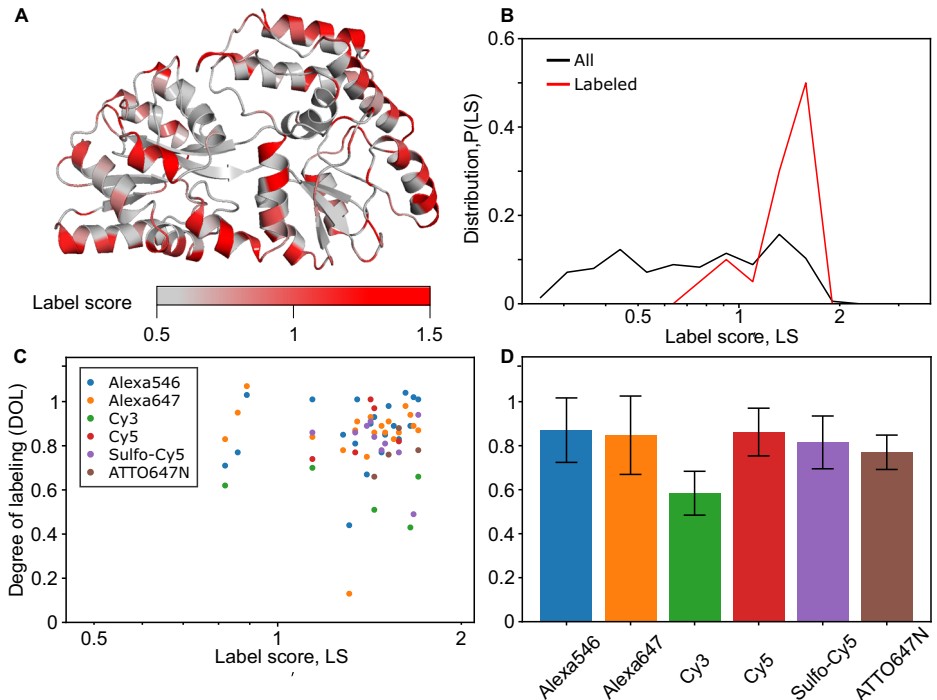

**Fig. 4 | Characterization of *LS* values with experimental parameters and degree of labeling (DOL) for single cysteine variants of MalE. A** Crystal structure of MalE in the apo state with *LS* color-coded. **B** *LS* distribution of apo MalE for all (gray) and the 20 successfully labeled residues (red). **C** *LS* vs. DOL for the MalE labeling data set, where different dyes are color-coded. **D** Average DOL for different dyes, based on the data in panel (**C**) with $n > 3$ for each dye. Error bars indicate the standard deviation. We do not observe a correlation of label efficiency and label score for the selected mutants with, in general, high label scores $LS > 1$. We do not observe a significant difference across fluorophore types ($p$-value > 0.05) except of Cy3, which showed significantly lower DOL values ($p$-value < 0.05) compared to all other fluorophores.

of the maltose binding protein (MalE) with single-cysteine labeling sites. MalE is a soluble bilobed protein with an open (apo) and a closed (holo) structure[61,73], which serves as the periplasmic component of the bacterial ABC importer MalFGK$_2$-E[74]. We visualized *LS* values for all sites of apo MalE in Fig. 4A and the corresponding distribution in Fig. 4B. The distribution shows an *LS* value range between 0 and 2; high values for *LS* appear mostly in positions of MalE near the surface when *LS* are mapped back to the structure (Fig. 4A).

First, we studied a data set of 20 variants of MalE, partially taken from previous work with (relatively) high *LS* scores representing residues that are good candidates for labeling according to our approach. For these experiments, we used the dyes Alexa546, Alexa647, Cy3, Cy5, sCy5, and ATTO647N and obtained an average degree of labeling (DOL) of 0.82 overall samples after protein labeling and SEC purification (Fig. 4). The *DOL* was determined using the molar ratio between fluorophore and protein concentration from the Lambert-Beer law: $DOL = c(\text{fluorophore})/c(\text{protein})$. All successfully labeled sites have an average of ~1.4, and almost 90% of them showed *LS* values > 1 (Fig. 4B, C). The distribution of the label scores for the successfully labeled sites is different from the distribution of all residues of the MalE protein (Fig. 4B), again confirming that *LS* provides valuable information about the suitability of protein residues to act as label sites. Our analysis shows, however, no correlation between *LS* and the experimentally determined DOL (Fig. 4C). This is not too surprising since all tested residues have relatively high label scores, and we focused on mutants with a reasonable chance of labeling and did not include measurements e.g., of buried residues with low label scores. Furthermore, we do not observe systematic differences between different dyes, suggesting that our method works robustly and is independent of the fluorophore (Fig. 4D and Supplementary Data; *LS* vs. DOL).

## Table 1 | Overview of expression and labeling properties of randomly selected MalE variants

| Cysteine variant | Control | Expression | Labeling |
|---|---|---|---|
| Q72C | positive | OK | OK |
| S211C | positive | OK | OK |
| K219C | positive | OK | OK |
| E309C | positive | OK | OK |
| E322C | positive | OK | OK |
| L7C | negative | reduced | X |
| W94C | negative | no expression | n.d. |
| I116C | negative | OK | X |
| G228C | negative | OK | OK |
| W230C | negative | OK | OK |

In a second set of experiments, we ranked all MalE residues by their label scores and then randomly selected 5 variants each from the best 10% *LS* scores (referred to as "positive control") and 5 residues from the worst 10% *LS* scores ("negative control"). For each of these 10 variants, we characterized the effect of the cysteine mutation in terms of the protein's expression yield and DOL using the dye sCy5 (all data are provided in the Supplementary Data Excel file and Table 1). All positive controls, i.e., MalE variants comprising residues with high label scores, expressed with high yields (> 15 mg from a 2 L expression culture) and could be labeled with a DOL > 85%. These findings again support that residues with high *LS* scores can be successfully expressed and labeled, in line with the first analysis of MalE point mutations (Fig. 4).

In contrast, two of the negative control variants showed reduced expression yield (L7C with 7.8 mg) or no expression at all (W94C).

Furthermore, two of the obtained four negative control variants showed DOL values < 2%. Interestingly, the other two negative control variants showed good expression yields and adequate DOL values, suggesting that not all residues with low *LS* scores are necessarily unsuitable for labeling. Taking all variants from this set of 10 MalE variants into account, there is a statistically significant correlation between *LS* score and DOL ($p = 0.03$ from a two-sample *t*-test), further supporting the approach presented here.

### Extension of the *LS* score to FRET experiments

To test our prediction tool for the design of a concrete biophysical assay, we extend it to FRET experiments. For this we combine the label score *LS* with an additional parameter for the rational design of FRET experiments. The central idea is to select residue pairs for FRET experiments that are (i) suitable as label sites based on *LS*, (ii) are separated by a distance that is close to the Förster radius of the dyes used (for maximum sensitivity) and (iii) that can detect conformational motion. Criteria i/ii are relevant to the case where one protein structure is available, and a residue pair is wanted with a distance close to the Förster radius of the dye pair. In this scenario, the researcher can use combinations of residues in different domains of the protein for maximal sensitivity. We define the FRET score *FS* of a residue pair *{i,j}* for a single protein structure as:

$$FS = \sqrt{LS_i LS_j} \cdot \left( 1 - 2 \left| \frac{1}{2} - E_{i,j} \right| \right), \tag{5}$$

*FS* considers the label scores $LS_i$ and $LS_j$ of two residues *i* and *j* in the protein structure with corresponding predicted FRET efficiency $E_{i,j}$ (see Supplementary Note 2 for details on the FRET efficiency prediction). *FS* is highest for residue pairs with predicted $E_{i,j} = 0.5$, i.e., an interdye distance similar to the Förster radius of the dye pair.

If two (interconverting) structures of a protein are available, one is interested to find FRET pairs that show the largest possible shifts in FRET efficiency. This scenario is encountered when ligand binding, protein-protein interactions, or other macromolecular interactions are studied and requires that distinct structures of the same protein, e.g., ligand-free and ligand-bound, are available. We define the FRET difference score $FS_\triangle$ of a residue pair *{i,j}* for two available structures *A* and *B* of the same protein as

$$FS_\triangle = \sqrt{LS_i^A LS_i^B} \sqrt{LS_j^A LS_j^B} \cdot |E_{i,j}^A - E_{i,j}^B|, \tag{6}$$

with the label scores *LS* of two residues *i* and *j* in two protein structures $A, B$ with their corresponding FRET efficiencies $E_{i,j}^A$ and $E_{i,j}^B$, respectively.

### Accessible volume calculations for FRET labels

To rationally establish a FRET assay with maximum sensitivity, it is necessary to operate at interprobe distances around the Förster radius. A crucial step for the calculation of both FRET scores is, therefore, the ability to predict interdye distances from the protein structures accurately (Fig. 5). The labelizer package supports three models for in silico fluorophore distance predictions. A rough approximation of expected FRET efficiencies can be obtained from the $C_\beta$ distances between two residues[55] (Fig. 5A). However, these distances can differ > 10 Å from the actual mean fluorophore positions, due to the size of the fluorophore and the flexible linkers (10–20 Å length) used for fluorophore attachment[75,76]. While distance changes are less impacted by such deviations, the absolute distances are significantly affected by the geometry of the labels (Fig. 5C). Neglecting these effects can reduce the sensitivity of a FRET assay by up to a factor of ~ 4 (Fig. 5D and Supplementary Figs. 7, 8).

To predict distances between fluorophore labels accurately, it is important to obtain accurate simulations of the accessible volumes (AVs) considering the size and shape of the dyes and their linkers. Molecular dynamics simulations have been successfully used for this purpose[77–79], yet they are too slow as a screening tool. Coarse-grained simulation via FRET-restrained positioning and screening system (FPS), where all positions on a grid are examined to decide whether it can be occupied by a fluorophore of the specified size and linker length, provide AVs that are in very good agreement with experimental values of interdye distances[49,76,80–86] (Fig. 5A). Comparing the calculated $C_\beta$-distances of the residues with FRET-averaged distances $R_{<E>}^{model}$ from AV simulations reveals deviations of 10 to 15 Å (RMSD, Fig. 5B and Supplementary Fig. 8A), highlighting the need to consider the dye and linker geometry. The computation time required for one pair of dyes using FPS, however, is still rather long for screening purposes, e.g., several hours when > 10.000 residue pairs should be considered (see Supplementary Table 9).

Therefore, we here introduce a simpler and faster distance estimator based on a spherical sector model (SSM) that estimates dye-accessible and dye-inaccessible volumes (Fig. 5B). SSM is used for screening purposes since it is 100 to 1000 times faster than currently available simulations such as FPS. Our algorithm relies on an approximation of the accessible volume by a spherical sector of angle $\alpha$ and radius *R* representing the linker length of the fluorophore (see Fig. 5C). The atoms of the protein within a radius *R* from the attachment site ($C_\beta$ atom) define an inaccessible volume (see Fig. 5B, C, pale red spheres). We find a direct relation between the center of mass of these atoms $\vec{d}'$ (inaccessible volume) and the center of mass of the accessible volume $\vec{d}$ (see Supplementary Note 2) as

$$\vec{d} = \left( 1 - \frac{3}{4} \frac{R}{|\vec{d}'|} \right) \vec{d}'. \tag{7}$$

We included a small correction $\varepsilon$ (~ 0.5 Å for typical fluorophores) to the linker length $\widetilde{R} = R + \varepsilon$ in this formula to compensate for the size of the fluorophore core (Supplementary Note 2 and Supplementary Fig. 7), and we used an estimation to convert the distance of the mean positions to FRET-averaged distances (Supplementary Note 2, Supplementary Fig. 8). To test our method, we performed distance simulations for 100 donor-acceptor pairs in 10 different protein structures, where we altered the linker length and the dye dimension with 35 variations resulting in 35.000 distance simulations in total. Our SSM approach gives results in good agreement with the FPS method with a deviation of ± 3 Å (RMSD, Fig. 5 and Supplementary Fig. 7), which is on the order of the intrinsic distance precision of FRET[76]. The mean-position distances are converted to FRET-averaged distances with an exponential correction factor at small distances (see Methods and Supplementary Fig. 8). The spherical sector method allows to screen > 10.000 FRET-pairs within seconds on a single CPU with < 1 ms calculation time per residue-pair (see Supplementary Table 9). Therefore, our standard settings are to use the SSM method for a first selection of suitable FRET-labeling positions and subsequently refine the best three hundred FRET pairs with the FPS AV-simulations[80,86]. Alternatively, our Python package allows calculating the $C_\beta$ distances (low accuracy) or the FPS-derived derived distances (long runtime) for all residues by manual selection.

### Experimental benchmarking of the FRET score

At first, we used the labelizer workflow to establish FRET assays for mechanistic studies of the ABC transporter-related prokaryotic substrate-binding protein PBP[59–61,87] (Fig. 6A). As seen in the crystal structures, PBP undergoes a ligand-induced transition from a ligand-free open (pdb: 1OIB, apo) to a ligand-bound closed state (pdb: 1PBP,

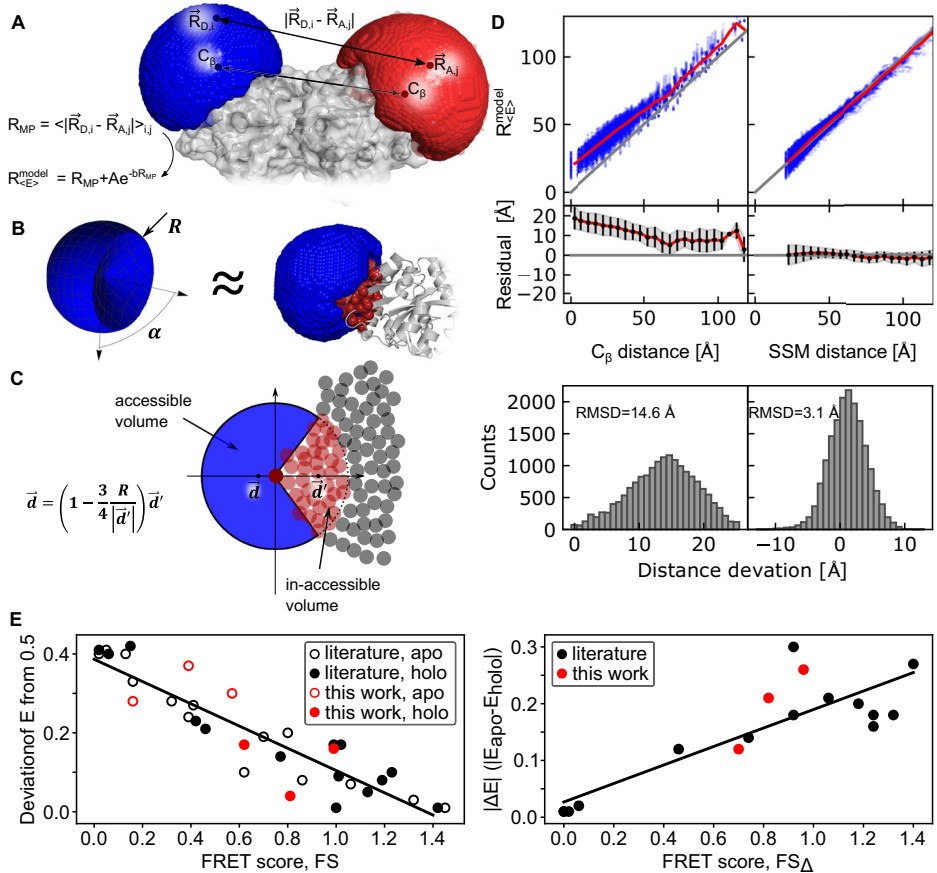

**Fig. 5 | Accurate prediction of interdye distances on proteins and experimental benchmarking of the FRET scores. A** Distance estimation with FPS computes a grid-based accessible volume to determine the mean position of the fluorophore $\langle \vec{R}_F \rangle$, the averaged inter-fluorophore distance $R_{MP} = \langle \vec{R}_D - \vec{R}_A \rangle$, and the efficiency-weighted average fluorophore distance $R_{\langle E \rangle}^{model}$ approximated with an exponential correction factor (see Supplementary Note 2). The illustration shows donor and acceptor labeling at residues S3C and P86C, respectively, in PBP (pdb: 1OIB). **B** Approximation of the accessible volume with a spherical sector. The spherical sector (left) is defined by the radius R (linker length of the fluorophore) plus an opening angle $\alpha$ and approximates the accessible volume simulated with FPS software (right, blue volume). The red spheres represent the protein atoms within radius R from the $C_\beta$ atom. **C** Illustration of the determination of the mean position in 2D. The circles represent the atoms of the protein. The inaccessible volumes are the atoms within a radius R (pale red circle) to the $C_\beta$ atom (dark red circle).

**D** Comparison of $C_\beta$ distances and modeled distances with the introduced spherical sector approximation compared to FPS-derived distances in 10 selected pdb structures with 35 different fluorophore parameters ($N = 32116$, top). We performed distance simulations for 100 donor-acceptor pairs in 10 different protein structures with the SSM approach, which is in good agreement with the FPS method with a deviation of $\pm 3$ Å (RMSD). The data $> 100$ Å are noisy due to low statistics. Histogram of the distance offsets from the $C_\beta$ atom for the distances in the range $40\,\text{Å} < R_{\langle E \rangle}^{model} < 75\,\text{Å}$ ($N = 17359$, bottom). **E** Analysis of the deviation of measured FRET efficiencies from 0.5 for selected MalE mutants taken from refs. 39,80. (Literature) and new experiments (this work) in apo (empty circle) and holo state (full circle) with respect to the computed FRET scores (left). Measured FRET efficiency shift between apo and holo of mutants in A plotted against the FRET difference score $FS_\Delta$ (right). Linear fits of the data are shown as solid lines with $R^2 = 0.84$ (left) and $R^2 = 0.73$ (right).

holo; Fig. 6A). Yet, the ligand binding mechanism of PBP, i.e., ligand-binding before conformational change (induced fit) or conformational change before ligand binding (conformational selection) has not been studied. Thus, our goal was to obtain assays with large changes in FRET efficiency upon the addition of the ligand inorganic phosphate for dye pairs with a Förster radius around 5 nm. We identified multiple suitable residue combinations with maximized positive and negative distance changes based on $FS_\Delta$ (Fig. 6B). We selected four double cysteine variants with large predicted shifts from long (low FRET) to shorter (higher FRET) distances upon phosphate binding. We selected those from the list of 300 refined pairs using the FPS parameters for Alexa Fluor 555-Alexa Fluor 647 (Fig. 6B and Supplementary Table 8). Before conducting FRET experiments, we characterized one of the double-cysteine variants PBP (S3C-I76G-P86C) and the cysteine-less PBP variant PBP (I76G) biochemically by ITC and obtained $K_d$-values of $10 \pm 5\,\mu M$ for PBP (S3C-I76G-P86C) and $19 \pm 6\,\mu M$ for PBP (I76G); mean $\pm$ SD from $n = 2$ protein preparations (Supplementary Fig. 9). These experiments suggest that protein labeling does not affect

substrate affinity. The I76G mutation was used in all PBP protein variants presented in this paper (Fig. 6 and Supplementary Fig. 9).

Subsequently, we labeled all four PBP variants using established procedures[39,88] (see Methods) and studied freely diffusing molecules with microsecond alternating laser excitation spectroscopy (μsALEX). For labeling, we used the donor-acceptor pair Alexa Fluor 555-Alexa Fluor 647 and the structurally related combination LD555-LD655 (Fig. 6D, E). The success of the labelizer prediction is seen in Fig. 6E and Supplementary Fig. 9, where high-quality smFRET histograms are obtained for all four PBP variants with low FRET in the apo (open conformation, no phosphate) and high FRET in the holo state (closed conformation, 480 μM phosphate). Analyzing the shift of the open to closed conformation and plotting the closed-state fraction as a function of ligand concentrations for PBP (S3C-I76G-P86C) with Alexa555-Alexa647 yields a $K_d$ of $16 \pm 6\,\mu M$ (Fig. 6F), which is in agreement with results for the unlabeled proteins (Fig. 6C). A similar behavior for the FRET-assay properties and biochemical characteristics are found for all PBP variants (Supplementary Fig. 10).

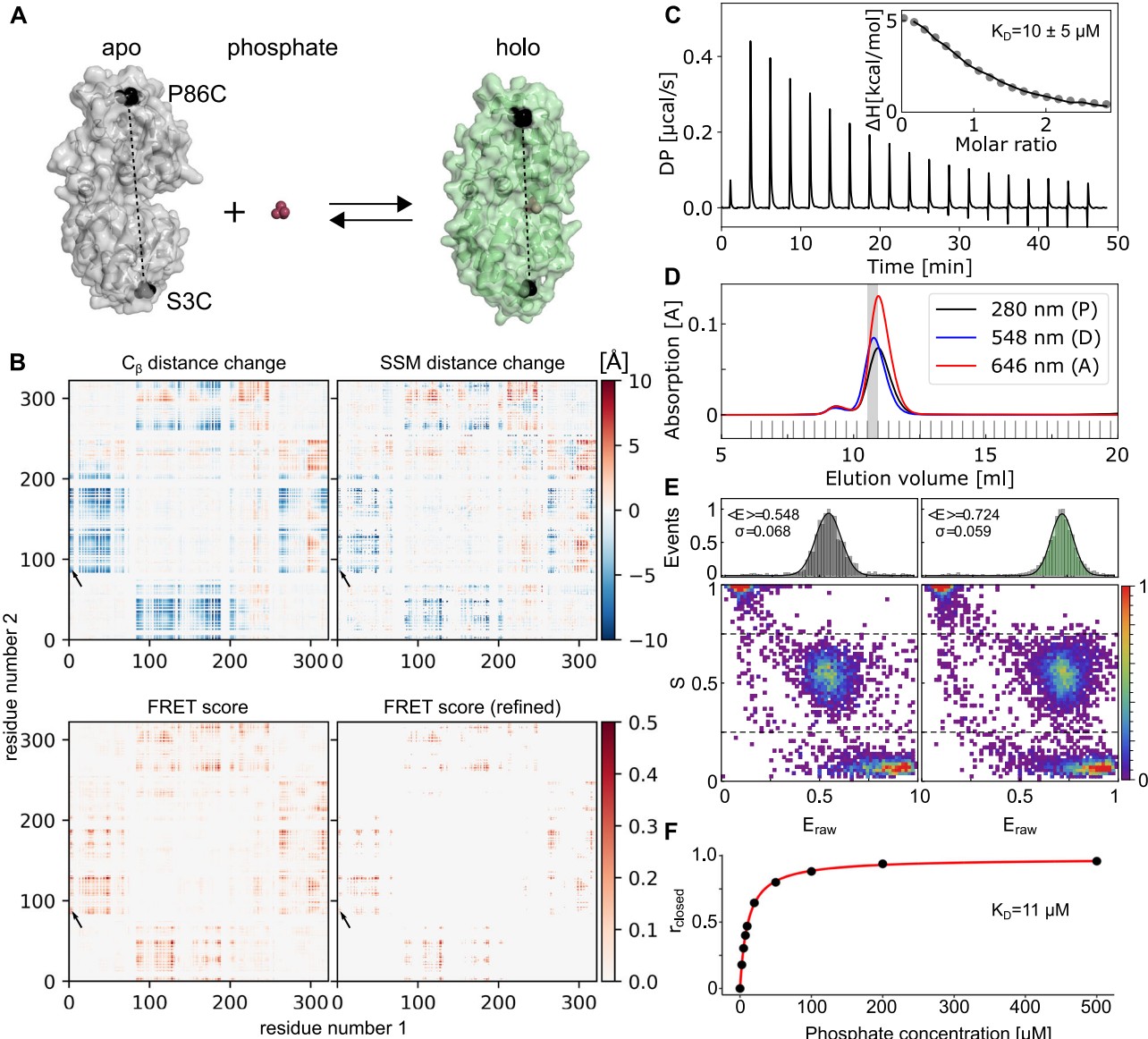

**Fig. 6 | Labelizer-based residue selection for FRET experiments and validation.**
**A** Crystal structure of PBP in the apo (gray, PDB-ID: 1OIB) and holo (green, PDB-ID: 1PBP) states with mutations S3C and P86C indicated in black. This variant of PBP also contains an I76G mutation that lowers the affinity for inorganic phosphate by ~ 200-fold compared to the wild-type protein. **B** Maps illustrating the distance change and associated FRET score for all pairs of mutants in PBP. The selected mutation S3C and P86C is marked with an arrow. The distance change of the attachment atom ($C_β$, top-left) and the change of the simulated spherical sector model (top-right) show a clear pattern of correlated movements. The calculated FRET score map of all pairs with average label score $LS > 1$ shows only a few spots (~ 4%) with promising FRET scores $FS > 0.2$ (bottom-left). The selection of the 300 pairs with the highest FRET score and refinement with FPS software (bottom-right) shows only minor variation compared to the screening map (bottom-left) for the analyzed data points. **C** Isothermal titration calorimetry (ITC) measurements of PBP (I76G) with average $K_d$-values. **D** Size exclusion chromatography of PBP with LD555-655 showing protein absorbance (280 nm) and fluorophore wavelength at 548 nm and 646 nm. **E** ES-FRET histograms of PBP (S3C-I76G-P86C) with LD555-655 in the ligand-free apo state (left) and with 480 μM phosphate (right). **F** Closed fraction ($r_{closed}$) as a function of the substrate concentration for PBP (S3C-I76G-P86C) with Alexa Fluor 555-647 determined from smFRET measurements. The red line is a fitted binding curve; three technical replicates gave an average $K_D$ of $10.8 \pm 0.2$ μM.

Beside the demonstration of the success of the labelizer procedure, these experiments provide so far unavailable information on the ligand binding mechanism of PBP. The lack of a pronounced closed-state population in the absence of ligand (Supplementary Fig. 9, apo) and the engulfed nature of the ligand in the closed state support the idea that PBP is likely to use a ligand binding mechanism like other structurally-related SBPs[60,61,73].

To go beyond a qualitative assessment of the labelizer routine, we analyzed a large pool of smFRET experiments of different MalE double-cysteine variants to quantitively benchmark the scores *FS* and $FS_Δ$. In detail, we analyzed 34 data sets of published[39,80] data and new data generated for this study (Supplementary Figs. 11, 12). For these the accurate FRET efficiencies of MalE in both apo- and holo-state are determined, including the respective interprobe-distances and their distance change upon maltose binding (Supplementary Data). This data set covers an experimental interprobe distance range from 3–7 nm for three distinct dye pairs with Förster radii of 5.1 nm (Alexa Fluor 555-Alexa Fluor 647), 5.8 nm (ATTO532-ATTO643) and 6.5 nm (Alexa Fluor 546-Alexa Fluor 647) and *E* values ranging from 0.2–0.9. An overview of experimentally determined and theoretical Förster radii is provided in Supplementary Fig. 11D.

Consistent with expectations, the calculated *FS* values correlate linearly with the difference of the experimentally determined mean FRET efficiency from 0.5 (Fig. 5E). Similarly, we observe a linear correlation between computed $FS_\Delta$ values and the experimentally observed change in FRET efficiency |$E_{holo}$-$E_{apo}$| upon ligand binding (Fig. 5E). Whereas pairs with large *FS* and $FS_\Delta$ values are desirable to detect changes upon ligand binding, pairs with high *FS* values of the two individual conformations, but $FS_\Delta \approx 0$ (MalE 84/352, Supplementary Fig. 12), can provide an important experimental control. Such pairs have a distance close to the Förster radius with (almost) no change in FRET efficiency upon conformational change. They can serve as negative controls to ensure that a protein or conformational changes do not influence fluorophores, e.g., via altered photophysics, lifetime and quantum yield changes, or for the characterization of quenchers such as metal ions[89], which can affect FRET efficiencies without conformational change.

Importantly, all analyzed fluorophore-labeled MalE variants used for smFRET had *LS* values > 1 and showed maltose affinities that are wildtype-like with $K_d$-values around ~1-2 μM (Supplementary Fig. 11). Taken together these analyses provide strong support for the idea that the *LS* is a useful indicator to identify residues that (i) allow fluorophore attachment, (ii) preserve protein function and in combination with FS (iii) enable systematic design of FRET assays.

## Discussion

Here, we present a general strategy to identify optimal residues for protein labeling using a naïve Bayes classifier. Based on a literature screening and bioinformatics analysis of 104 proteins with 396 successfully labeled residues, we identified a set of four parameters, which we combined into a label score to quantitatively rank residues according to their suitability as label sites. We show, using data from the literature and new complementary experiments, the predictive power of this labeling score and extend the method to systematically select residue pairs for FRET experiments, which we believe can be extended at a later stage to consider the specific properties of the label and also other biophysical assays beyond FRET.

To widely disseminate our methodology, we provide a Python package called "labelizer", which implements the analysis of the pdb-structure, label score calculation, and FRET assay scoring. The labelizer analysis routine can be modified and extended, to accommodate specific research questions and to build upon the work presented here. To make the methodology widely available to non-expert users, all key functionalities are available as a web server with an intuitive and user-friendly interface https://labelizer.org. The web server supports the label score calculation and its use for FRET experiments with default parameters for the most frequently used fluorophores. For this purpose, pdb-files can be loaded automatically and preprocessed from the pdb-database. We further retrieve conservation scores directly from an independent installation of the ConSurf server[62,63] without the need of uploading any information (except when modified or user-specific pdb files should be used). The web server visualizes the different scores in an interactive 3D structure viewer and provides a table with filter options for customized restrictions upon residue selection. Furthermore, human-readable result files (csv, json) are available for subsequent analysis. With the developed method, we hope to provide scientists in various research fields (biochemistry, molecular biology, bioimaging, high-resolution optical microscopy, and single-molecule biophysics) with a tool that enables them to systematically design assays and justify the residue selection.

A challenging aspect of our analysis is the final selection of residues by the user based on the labelizer output. Since this step is decisive for which residues are used in experiments, the selection goes hand in hand with an assessment and interpretation of the *LS*/FS value distributions of the analyzed protein. It is difficult to define clear threshold values for residues to be excluded based on *LS*/FS, yet our findings empirically suggest that residues with *LS* values < 1 are less likely to be useful in experiments. Since the FRET-score values additionally depend on the underlying *LS* distribution, it is difficult to give general recommendations. We stress that the user of the algorithm should inspect the specific *LS*/FS distributions for each protein. For the residues ranked highest, we recommend the user to verify this selection with prior (expert) knowledge on the protein. A key question would be whether the highly-ranked residues, i.e., those favored by the labelizer, are known to negatively impact secondary structure, ligand-binding, biomolecular interactions, or protein folding. Additional information might also come from other biophysical approaches such as CD spectroscopy, FTIR, MD simulations or EPR studies, considering any information that can help to assess if key residues, which should not be altered, are actually (falsely) suggested by our algorithm.

An interesting future direction for further development of the labelizer is to include more parameters (e.g., also fluorophore-dependent ones) with a potential differentiation of residues based on the selected fluorophores related to the specific charge environment on the protein or proximity to specific amino acids, e.g., tryptophane or histidine. We also plan to combine different parameter scores to improve the predictive ability of the labelizer, which might happen within one category, e.g., via simultaneous use of half-sphere exposure (HSE) and relative surface area (RSA) to combine the amino-acid direction and surface area or between categories, e.g., solvent exposure and cysteine resemblance. Furthermore, normal mode analysis (e.g., NMSim webserver[90,91]), mutation-specific energy analysis (e.g., SDM[92,93]), or tailored MD-simulations[94] could be used to identify FRET-residue pairs for analysis of conformational motion when only one protein structure is available. The concept of FRET scores could be also extended towards other fluorescence assay types related to fluorophore quenching[95,96], protein-induced fluorescence enhancement[97,98], and others[99,100]. We also envision applying the labelizer approach in related applications, such as EPR-distance measurements, since the methods share similar requirements in regard to residue selection[37–39].

Another direction for future improvement and extension of the database and the algorithm would be to revise the available *PS* values by an extended database, where particularly positions with low or no yield of labeling, could be an important new class of information. Such an improved training data set can be obtained via a feedback loop, where researchers supply information on successfully and unsuccessfully labeled residues via a form planned on our website. Unsuccessful results are of particular interest since negative results are rarely found in the literature (mainly successful results are published), and we were not able to collect enough negative examples from researchers directly. Therefore, we call on the scientific community to use the labelizer and to provide feedback on the approach and on positive and negative results, where labeling of specific residues was successful or failed, respectively. Finally, once a much larger dataset of labeled and non-labeled residues is available, applications of other machine learning procedures (e.g., support vector machine or neural networks) could significantly enhance the predictions.

## Methods
### Database generation
To identify parameters with predictive power for the possibility to label residues in proteins, we created a dataset based on a non-automated screening of more than 1000 publications published or preprinted, which were available on or before December 2020 with a focus on the field of single-molecule microscopy and single-molecule FRET. The papers were screened to identify proteins and residues that were labeled successfully with a fluorophore and that satisfied the following criteria: (i) the proteins had a structure available in the PDB database (with PDB identification code); (ii) the protein was labeled via site-specific mutagenesis and introduction of cysteines or UAAs; (iii)

the protein was successfully labeled synthetic organic fluorophores (or spin labels) and used preferentially single-molecule assays. In order to increase the number of database entries, we complemented our search whenever some information was missing. Typical cases were missing PDB identification codes or residue numbers. In this case, the required information was obtained from other referenced papers (often) of the same research group.

For each successfully labeled protein variant, which fulfilled the aforementioned criteria, the following information was collected:

- Protein (PDB identification code)
- Soluble or membrane protein
- Stoichiometry (monomers, dimer, complexes)
- Homology model (true/false)
- Labeled residue (chain and residue number)
- Mutation (cysteine or UAA)
- Assay type (smFRET, imaging, bulk-FRET, other)
- Name of labeled fluorophores
- Research group
- Publication reference

Additional notes were gathered to account for issues such as: (i) dimer and polymer protein structures, which were crystallization artefacts and needed to be deleted for structural analysis; (ii) missing residues in protein structure, i.e., when parts of the protein were not resolved completely; (iii) we identified inconsistencies or missing information. The final database with information on those positions in proteins that were successfully labeled had 396 successfully labeled residues in 112 different chains in 104 different protein structures (Supplementary Data). As a comparison, we used a representative set of proteins (PDBselect, November 2017)[67,68] as a random reference database to check how representative the analyzed pdb structures are. Therefore, we randomly selected 300 chains (out of 4184 chains) from the PDBselect database and performed the identical analysis with those pdb files. This important comparison shows that the selection of labeled proteins and residues is representative of the pdf content, indicated by only minor deviations between both P(s) distributions, mostly within statistical errors (see Supplementary Fig. 2).

**Parameter frequency calculation.** For every extracted parameter, the relative frequency defines a parameter score

$$PS = \frac{P(s|l)}{P(s)}, \tag{8}$$

where $P(s)$ is the probability distribution of the score $s$ (calculated from the 112 chains of the database) and $P(s|l)$ is the probability distribution of the score given that the residue was labeled (calculated from the 396 successfully labeled residues).

The error bars $\sigma_{sl}$ and $\sigma_s$ for $P(s|l)$ and $P(s)$, respectively, were determined from Poissonian counting statistics as $\sigma_{sl} = \sqrt{P(s|l)/n}$ and $\sigma_s = \sqrt{P(s)/n}$ with $n$ being the total number of evaluated residues. The error bar $\sigma_{PS}$ for $PS$ follows from standard error propagation rules:

$$\sigma_{PS} = \sqrt{\frac{\sigma_{sl}^2}{P(s|l)^2} + \frac{\sigma_s^2}{P(s)^2}} PS. \tag{9}$$

**Parameter information analysis.** To evaluate the amount of information a single parameter score inheres, we used three measures to estimate the deviation from an equal distribution, which corresponds to the case of zero information.

We used standard Pearson correlation for a pair of numeric parameters

$$MSD(PS) = \frac{\sum_{i=1}^{n}(PS(i)-1)^2}{n} \tag{10}$$

with n the number of bins/categories.

We used standard Pearson correlation for a pair of numeric parameters

$$gini(PS) = \frac{\frac{n-1}{2}\sum_{i=1}^{n}PS(i) - \sum_{i=2}^{n}\sum_{j=1}^{i-1}PS(j)}{\frac{n}{2}\sum_{i=1}^{n}PS(i)} \tag{11}$$

with n the number of bins/categories.

We used an adapted Shannon entropy accounting for the number of bins/categories as

$$H(PS) = \frac{-\sum_{i=1}^{n}\widetilde{PS}(i)\ln\left(\widetilde{PS}(i)\right)}{\ln(n)} \tag{12}$$

with a normalized parameter score $\widetilde{PS}(i) = PS(i)/\left(\sum_{j=1}^{n}PS(j)\right)$ and n the number of bins/categories.

**Parameter cross-correlation.** To evaluate the mutual statistical dependence of all calculated parameters, we use three different types of correlation coefficients, depending on the datatypes of the parameters:

We used standard Pearson correlation for a pair of numeric parameters

$$r_{NN} = \frac{\sum_{i=1}^{n}(x_i-\bar{x})(y_i-\bar{y})}{\sqrt{\sum_{i=1}^{n}\left(x_i-\bar{x}\right)^2}\sqrt{\sum_{i=1}^{n}\left(y_i-\bar{y}\right)^2}}, \tag{13}$$

with $n$ different residues with parameter scores $x_i, y_i$ and corresponding mean values $\bar{x} = 1/n\sum_{i=1}^{n}x_i$ (and $\bar{y}$ accordingly)[101].

We used the interclass correlation for a pair of a categorical parameter and a numeric parameter[102]. The $n$ data points are grouped in k categories $c_i$ with $i \in \{1, 2, \ldots, k\}$ of length $n_i$.

$$r_{CN} = \frac{MST - MSE}{MST + (n_0 - 1)MSE}, \tag{14}$$

with

$$MST = \frac{\sum_{i=1}^{k}n_i\sum_{j=1}^{n_i}\left(\bar{x}_i-\bar{x}\right)^2}{k-1}, \tag{15}$$

$$MSE = \frac{\sum_{i=1}^{k}\sum_{j=1}^{n_i}\left(x_{i,j}-\bar{x}_i\right)^2}{n-k}, \tag{16}$$

$$n_0 = \frac{n - \sum_{i=1}^{k}n_i^2/n}{k-1}, \tag{17}$$

where $\bar{x}_i$ is the mean of category $i$, $\bar{x}$ the mean of all data, $x_{i,j}$ the $j^{th}$ numeric value in category $c_i$, and $(n_0 - 1)$ the averaged interclass degree of freedom[102].

We used Cramer's V for a pair of categorical parameters[103]. The data are grouped in the two categories $c_i$ with $i \in \{1, 2, \ldots, k\}$ and $d_j$ with $j \in \{1, 2, \ldots, l\}$.

$$r_{CC} = \sqrt{\frac{\chi^2}{n(\min(k,l)-1)}}, \tag{18}$$

with

$$\chi^2 = \sum_{i=1}^{k}\sum_{j=1}^{l}\frac{\left(n_{i,j}-\widetilde{n}_{i,j}\right)^2}{\widetilde{n}_{i,j}}, \tag{19}$$

where $\widetilde{n}_{i,j} = (\sum_{j=1}^{l}n_{i,j})(\sum_{i=1}^{k}n_{i,j})/n$, n total number of residues and $n_{i,j}$ number of residues of class $c_i$ and $d_j$. The cross-correlation was

calculated for every combination of the 28 extracted parameters to identify dependencies, as shown in Fig. 2.

**Parameter selection criteria.** The selection of a suitable parameter set is based on two criteria. First, a joined correlation for any combination of parameters is calculated as

$$r_{set} = \sqrt{\sum_{i=1}^{n} r_{ij}}, \tag{20}$$

with $r_{ij}$ the correlation of parameter $i$ with $j$ and $n$ the number of selected parameters (in our case, 4). Secondly, we used three measures to characterize our parameter sets:

We calculate the t value of the calculated label scores as

$$t = \frac{\mu_l - \mu_{all}}{\sqrt{SEM_l^2 + SEM_{alll}^2}} \tag{21}$$

with the mean values $\mu_l$, $\mu_{all}$ and standard error of the mean $SEM_l$, $SEM_{all}$ of the labeled/all residues, respectively.

The dynamic range was calculated as the standard deviation of the logarithmic values $\sigma(\log(LS_{all}))$.

The suppression/enhancement of the labeling score of labeled residues for small/large values was calculated from the slope of a linear least square fit to the logarithm of the label score $LS$ and the label score distribution of labeled residues and all residues. The data are binned into logarithmic bins with bin intervals $\left[1.5^i, 1.5^{i+1}\right]$ for $i \in \{-12, \ldots, 11\}$ and fitted to the function

$$\log\left(\frac{P(LS|l)}{P(LS)}\right) = m \log(LS) + \log(c) \tag{22}$$

where $LS$ is the label score and $P(LS)/P(LS|l)$ the probability distributions of the label score of all and the labeled residues. The slope $m$ is used as analysis parameter form the fitted values $m, c$.

## Statistics & reproducibility
No statistical method was used to predetermine sample size. No data were excluded from the analyses. The experiments were not randomized except for new MalE mutants added during the revision of this paper. The investigators were not blinded to allocation during experiments and outcome assessment.

## Protein production and labeling
In the current study, we used single cysteine variants of MalE (Fig. 4) that were obtained and fluorophore-labeled according to published procedures[60,61]. PBP double cysteine variants were produced for this study. The coding sequence for the *E. coli* K12 *phoS* gene (Genbank coding sequence NC_000913.3, 3910485 - 3911525 complement, protein accession number NP_418184.1), with amino-acid changes (A17C and A197C) corresponding to the rho-PBP fluorescent biosensor variant[104] was synthesized (Invitrogen GeneArt Gene Synthesis, Thermo Fisher) without its N-terminal signal sequence (25 amino acid N-terminal deletion). This construct utilized flanking NdeI/XhoI sites and was subcloned into the pET20b expression vector. The resulting construct encoded a C-terminal His-tag fusion. The S3C-P86C-PBP mutant, with the additional I76G mutation that reduces the wild-type affinity ($K_d$ 0.07 μM) of the protein for inorganic phosphate by ~200-fold[87] was created using a protocol based on the Stratagene Quikchange protocol. As a control, a variant was also created with only the I76G mutation.

*E. coli* BL21 (DE3) *pLysS* cells transformed with the S3C-P86C-PBP mutant expression plasmid (or the plasmid for the control variant) were used to inoculate Terrific Broth (TB; Carl Roth, Karlsruhe,

Germany) supplemented with 100 μg/ml carbenicillin (Carl Roth) and 0.2% glucose to an optical density at 600 nm ($OD_{600}$) of 0.1 AU at 37 °C with shaking at 200 rpm. At an $OD_{600}$ of ~0.3 AU, isopropyl b-D-1-thiogalactopyranoside (IPTG, Carl Roth) was added to a final concentration of 0.5 mM, followed by ~24 h incubation. Cells were harvested by centrifugation (5000 × g, 20 min, 4 °C) at a final culture $OD_{600}$ of 3-4 AU, resuspended in 35 ml 20 mM HEPES pH 7.5, 300 mM NaCl, 10% glycerol containing protease inhibitor (cOmplete, EDTA-free Protease Inhibitor Tablets, Sigma; 1 tablet/50 ml solution), and frozen and stored at − 80 °C.

The resulting cell suspension was thawed, supplemented with 5 mM β-mercaptoethanol (β-ME) and 10 mM imidazole (Carl Roth), and then sonicated (Branson Digital Sonifier 450, Danbury, CT, USA) on ice for 10 min (Amplitude, 25%; 0.5 sec on and 0.5 sec off). Insoluble fractions containing cell debris were separated by centrifugation (165,000 × g for 1 h at 4 °C). The soluble fraction was incubated with 1.5 ml of Ni Sepharose 6 Fast Flow resin (GE Healthcare) for 1 h at 4 °C. The resin with bound protein was then washed with 80 ml of buffer containing 25 mM imidazole. Bound protein was eluted in 10 ml buffer with 500 mM imidazole. The elution fraction was concentrated to < 0.5 ml using a Viva Spin 20 concentrator with a 10 kDa MWCO (Th. Geyer, Renningen, Germany), and subjected to further purification by size-exclusion chromatography (SEC; using ÄKTA pure system, and Superdex 75 Increase 10/300 GL column (GE Healthcare)) in 20 mM Tris-HCl pH 8.0, 100 mM NaCl, 10 mM imidazole. The final purified proteins were >95% pure as assessed by sodium dodecyl sulfate polyacrylamide gel electrophoresis (SDS-PAGE).

His-tagged MalE and S3C-P86C-PBP proteins were labeled according to published procedures[60,61]. The proteins were incubated with 1 mM DTT to reduce cysteine residues. Following dilution to lower the DTT concentration to <0.05 mM (so as not to interfere with the binding of protein to the metal-affinity resin), the proteins were immobilized on 200 μl of Ni Sepharose resin. The resin was then washed with 12 ml of 50 mM Tris-HCl pH 7.4-8.0, 50 mM KCl, 5% glycerol for MalE and SBD2 (Buffer A), and 20 mM Tris-HCl pH 8.0, 100 mM NaCl, 10 mM imidazole for PBP. 28 nmoles of PBP were then incubated overnight with 50 nmol of each fluorophore dissolved in 2 ml of the appropriate buffer. An unreacted fluorophore for MalE and SBD2 was removed by washing the resin with 12 ml of Buffer A followed by 12 ml of Buffer A containing 50% glycerol. For PBP, a single 12 ml wash was performed. Bound MalE and SBD2 were eluted with 0.5 ml of Buffer A containing 500 mM imidazole, whereas PBP was eluted with 1 ml of buffer with 500 mM imidazole. The labeled proteins were further purified by size-exclusion chromatography (using ÄKTA pure system, and Superdex 75 Increase 10/300 GL column (GE Healthcare)). The absorbance of protein (280 nm) and fluorophore (532 and 640 nm) was used for the determination of molar concentrations in samples and labeling efficiency, i.e., [Fluorophore]/[protein]*100.

## Affinity measurements: Isothermal titration calorimetry and MST
Binding affinities of I76G-PBP and unlabeled S3C-P86C-PBP for inorganic phosphate were determined with a MicroCal PEAQ-ITC microcalorimeter (Malvern Panalytical) at 25 °C. Protein from a diluted solution was concentrated to ~30 μM using a Viva Spin 6 concentrator with a 10 kDa MWCO. The filtrate was used to prepare the phosphate solution at 450 μM. The reaction cell was filled with the protein solution and titrated in 19 steps of 2 μl each of phosphate solution in 160 s intervals. A baseline control was obtained from measurements made with protein-free filtrate in the reaction cell, and this baseline was subtracted from the experimental thermograms. Data were fitted to a single binding site model using the Setup MicroCal PEAQ-ITC Analysis Software provided by the manufacturer.

## smFRET spectroscopy and data analysis

smFRET experiments of PBP and MalE were carried out on a home-built ALEX setup[60,88]: PBP was studied by diluting the labeled protein to concentrations of $\approx 80$ pM in a 100 μl drop of buffer (20 mM Tris-HCl pH 8.0, 100 mM NaCl, 10 mM imidazole) on a coverslip supplemented with the ligand phosphate as described in the text and figures. Before each experiment, the coverslip was passivated for 3 minutes with a 1 mg/ml BSA solution in the buffer. The measurements were performed without a photostabilizer. The fluorescent donor molecules were excited by a diode laser at 532 nm operated at 60 μW at the sample in alternation mode (50 μs alternating excitation and a 100 μs alternation period). The fluorescent acceptor molecules were excited by a diode laser at 640 nm operated at 25 μW at the sample. Data analysis was performed using a home-written software package as described in reference[60]. Single-molecule events were identified using an all-photon-burst-search algorithm with a threshold of 15, a time window of 500 μs, and a minimum total photon number of 150[105]. E-histograms of double-labeled FRET species with LD555 and LD655 were extracted by selecting $0.25 < S < 0.75$. E-histograms of the open state without ligand (apo) and closed state with saturation of the ligand (holo) were fitted with a Gaussian distribution $A\,e^{-\frac{(E-\mu)^2}{2\sigma^2}}$.

## Reporting summary

Further information on research design is available in the Nature Portfolio Reporting Summary linked to this article.

## Data availability

Primary research data generated in this study have been deposited in zenodo under accession code https://zenodo.org/records/14965046.

## Code availability

The web server with an intuitive user interface and default analysis settings is available under https://labelizer.org. The software is available as a Python package "labelizer" as source code under https://github.com/ChristianGebhardt/labelizer. The databases and additional information can be accessed from https://github.com/ChristianGebhardt/labelizer-supplement or from the online version of the paper.

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

## Acknowledgements

This work was financed by an ERC Starting Grant (ERC-StG 638536 - SM-IMPORT to T.C.) and an ERC Consolidator Grant (ProForce to J.L.), Deutsche Forschungsgemeinschaft (GRK2062 project C03 to T.C., SFB863 projects A11 and A13 to J.L. and T.C.; Sachbeihilfe CO879/4-1 to T.C.), BMBF (KMU innovative "quantum FRET" to T.C.) LMU excellent, the Center for Integrated Protein Science Munich (CiPSM), and the Center for Nanoscience (CeNS). We thank all members of the Cordes lab for actively testing the labelizer procedure and web server, in particular Rebecca Mächtel, Alessandra Narducci, Oliver Brix, Leonor Correia, Shirsha Roy, and Chuyu Han. We finally thank our colleagues Gregor Hagelücken, Eitan Lerner, Nicole Robb and Giorgos Gouridis for discussions and support of the project.

## Author contributions

C.G. and T.C. conceived and designed the study. C.G. performed research, data analysis, and software implementation. J.L. provided analytical tools. K.S. and T.C. analyzed data. C.G., P.B., R.S., and K.S. implemented the webserver. A.K.S., N.W., and G.G.M.M. performed research. D.A.G. prepared PBP variants, performed research, and analyzed data. J.L. and T.C. supervised the study and acquired funding. C.G., J.L., and T.C. discussed and interpreted the results and wrote the manuscript in consultation with all authors.

## Funding

## Competing interests

The authors declare no competing interests.
