## [Transparent Peer Review file · Nature Communications]

Labelizer: systematic selection of protein residues for covalent fluorophore labeling

Corresponding Author: Professor Thorben Cordes

Version 0:

Reviewer comments:

Reviewer #1

(Remarks to the Author)

In this manuscript, the authors developed a systematic and automated approach called Labelizer for selecting suitable labeling sites in proteins. The approach uses a naïve Bayes classifier to identify optimal residues for protein labeling. The authors also developed a python package and a webserver to make the method available to a large community of researchers and to build up a central open-access database of labeled protein residues to continuously improve and refine the Labelizer approach. This paper provides a valuable tool for researchers working with macromolecules, particularly proteins, and has the potential to advance research in various fields. However, I have a few concerns about the database and the calculation of the labeling score:

1. In many cases, the labeled residues reported in literature were chosen because their positions in the 3D structure could reflect the conformational changes relevant to function, in addition to considering the ease of labeling. Is there any consideration of dealing with this bias of the data?
2. On page 5, the authors just omit the $P(l)$ term because its "harder to access". The authors mentioned that " $P(l)$ only scales the final probability". However, as long as $P(l)$ is not a uniform distribution, it affects the labeling score. The simplest model of $P(l)$ may be the chemical nature of the residues, i.e., cysteine has a distinct core form other residues.

Reviewer #2

(Remarks to the Author)

In this work, the authors detail the creation of Labelizer, a tool for choosing fluorescent labels on proteins. They mined publications (which are nicely indexed in a database) and identified multiple metrics important when considering choice of label position. They then combine the sequence and abstracted structure information to score residues for positioning donor and acceptor fluorophores. They perform experiments to show that their predictions are accurate.

The authors reference the current published literature appropriately.

The manuscript was well written, clear to understand, and thorough as to their motivation, previous literature, and techniques used. I additionally appreciated the inclusion of the mined database as well as the effort to make this reachable/useable for the community through a webserver. This is really an excellent job of streamlining or applying structural biology to a problem. My largest concern in this work is that there's already a confirmation bias at work in this parameterization, since the only information to draw from are reported labels that have been published and, therefore, have "worked." To test the robustness of the methodology, it would be useful to have a negative control or to include data from tags which have not worked in a way where it can be used to strengthen predictions (ex., how protein force fields are parameterized to capture minima AND barriers).

Additionally, I think the authors should expand the discussion to report on the quality of the information conveyed by a specific label, and not just the ability of the label to be covalently attached. Should part of this guidance speak to the success of the information you get out of the label (is it fit for purpose?), instead of only the success of labeling the protein? I guess the argument could be that this provides a robust mechanism for making suggestions, and the researcher needs to make the final decision based on what they are interested in measuring, but the ability to do the latter negates the need for the former. This is touched on as point (iii) on page 10, but should be expanded on in the discussion of FRET benchmarking and algorithm development.

Minor points:

Figure 1A should be boxed in to separate and emphasize the parts of the Labelizer procedure in B and C. The A panel is a

summary of problems labeling may introduce into assays – perhaps boxing it in and labeling it as such would make the figure clearer. Fluorescence properties should be properties. Graphs in Panel C, parts 2 and 3 should have labeled axes (I am assuming 2 is residue # and I do not know what 3 is – probability distribution along residues?)
Supplementary Figure 1A should read “Labeled Residues per protein” on the X-axis. Solubility should be solubility.
Supplementary Figure S8 should have Angstroms instead of A.

Reviewer #3

(Remarks to the Author)

This manuscript “Labelizer: systematic selection of protein residues for covalent fluorophore labeling” by Gebhardt, C. et al has described a general and, as claimed, quantitative strategy to identify optimal residues for protein labeling using an approach called naïve Bayes classifier. As described, the manuscript analyzed available literature and bioinformatics of more than 100 proteins with about 400 successfully labeled residues, from which the authors claimed that this strategy can be used to determine the labeling score to rank residues for their suitability as a labeling site. The approach was then expanded to a pair of residues for the systematic selection of a residue pair for FRET experiments. Finally, a Python package, which they call “labelizer”, is developed to make the method available to a larger research community with anticipation of building a central open-access database for both successfully and non-successfully labeled proteins/residues. An algorithm to determine the fluorophore(s) labeling site with little to no effect on protein folding and function is definitely a very desirable tool in the fluorescence/FRET community. Figuring out the labeling sites at a distance of Förster radius in a protein is another sensitive tool for studying protein-ligand and other biomolecular interactions that encompasses small to large conformational changes of proteins. Overall, I find this manuscript interesting and well-written and I believe that it will be a very helpful resource for those who are seeking protein labeling. However, I also noted some issues (as outlined below) that need to be addressed.

- 1) Provide a reference for ‘Förster Radius’ in this part of the sentence “....but that are also compatible with the assay requirements, e.g., for FRET to result in an inter-fluorophore distance close to the Förster Radius R_0 .”
- 2) What does the –ve value of the conservation score in Fig 2A represent?
- 3) Based on the discussion related to Fig 2, it is my understanding that the low correlation parameters are preferred among different parameters (CS, SE, CR, and SS) to achieve successful labeling. However, the manuscript does not clarify why this is the case.
- 4) The parameters ‘ ρ ’ and ‘ Π ’ in equation 3 are not defined.
- 5) To help readers better understand the manuscript, the authors should discuss the physical meaning of the critical parameters. For example, what does the label score value of 1 mean in Figure 3? Similarly, what does the FRET Score (FS) value of 1 mean in Fig 6? This applies to other parameters as well. This information will be a reference point for readers to understand the data.
- 6) What does “Label (x100)” represent in Fig. 3C?
- 7) Panel E, Fig 6, y-axis label: a space seems to be missing between “Deviation” and “of”.
- 8) On Page 10, in the paragraph “Our analysis shows, however, no correlation between LS and the experimentally determined DOL (Figure 4C). This is likely since all residues tested have relatively high label scores and we focused on mutants with a reasonable chance of labeling and did not include measurements e.g., of buried residues with low label scores.” I believe that an actual correlation should be observed to claim that the prediction tool actually works. The explanation provided is not satisfactory to support the claim. At least, I found that the claim is not easy to understand.
- 9) Data in Fig. 6F seems incomplete as the first concentration of phosphate used already shows the rclosed value of ~0.5. Can the authors try a couple of lower phosphate concentrations and update the graph? Also, is the rclosed value determined to be zero via analysis (or experiment) in the absence of phosphate or this is an assumption? This has to be clarified.
- 10) The manuscript does not explain some of the important observations and I suggest that they should not be left uninterpreted. For example, Fig. 4 panel D, why the Cy3 labeling score is significantly lower as compared to all other fluorophores. Similarly, in Fig. 3, panel D, the sudden downward trend (red fit after around 100 Å) seems rather unusual given the upward trend of the raw data. This needs to be explained in the figure legend.
- 11) Understandably the developed tool “labelizer” utilizes careful use of parameters and multiple intertwined analyses with boundaries and so on. This is something hard to fully assess by reviewers. The authors should make every effort to ensure that there are no errors in the tool and that what it predicts is fully reliable.

Version 1:

Reviewer comments:

Reviewer #1

(Remarks to the Author)

The authors have addressed my concerns.

(Remarks on code availability)

Reviewer #2

(Remarks to the Author)

The authors have addressed all of my concerns. I appreciate the diligence of the expanded MalE study, which is very convincing for the enrichment for labeling that one gets from using the Labelizer approach.

(Remarks on code availability)

Reviewer #3

(Remarks to the Author)

My comments are addressed and the manuscript is much improved after revision.

(Remarks on code availability)

I have limited expertise on code and therefore not able to comment.

Reviewer #1 (Remarks to the Author):

In this manuscript, the authors developed a systematic and automated approach called Labelizer for selecting suitable labeling sites in proteins. The approach uses a naïve Bayes classifier to identify optimal residues for protein labeling. The authors also developed a python package and a webserver to make the method available to a large community of researchers and to build up a central open-access database of labeled protein residues to continuously improve and refine the Labelizer approach. This paper provides a valuable tool for researchers working with macromolecules, particularly proteins, and has the potential to advance research in various fields. However, I have a few concerns about the database and the calculation of the labeling score:

Reply: We thank the reviewer for carefully reading the manuscript, the overall positive assessment, and for providing suggestions to improve the paper.

1. In many cases, the labeled residues reported in literature were chosen because their positions in the 3D structure could reflect the conformational changes relevant to function, in addition to considering the ease of labeling. Is there any consideration of dealing with this bias of the data?

Reply: We fully agree with the referee that the literature underlying our data set is biased by the fact that researchers have reported on residues that could be labeled and successfully report on relevant conformational changes. We now explicitly point this out in the revised main text. This bias was also an important factor in our decision to use conditional probabilities for scoring the residues.

2. On page 5, the authors just omit the $P(I)$ term because its "harder to access". The authors mentioned that " $P(I)$ only scales the final probability". However, as long as $P(I)$ is not a uniform distribution, it affects the labeling score. The simplest model of $P(I)$ may be the chemical nature of the residues, i.e., cysteine has a distinct core form other residues.

Reply: While the referee is correct that $P(I)$ affects the numerical value of the labeling score, i.e., it scales the distribution, it does not affect the relative predictions of which residues are suitable for labeling. We now state this aspect more clearly in the main text of the revised manuscript. We do not believe that $P(I)$ can easily be computed from e.g., the fact that a residue is cysteine. Again, this is not a problem since we explicitly include the identity of each residues and this information is included in the conditional probability $P(s|I)$.

Both comments 1./2 of referee #1 resulted in changes in the main text between eqn. 1 and 2.

Reviewer #2 (Remarks to the Author):

In this work, the authors detail the creation of Labelizer, a tool for choosing fluorescent labels on proteins. They mined publications (which are nicely indexed in a database) and identified multiple metrics important when considering choice of label position. They then combine the sequence and abstracted structure information to score residues for positioning donor and acceptor fluorophores. They perform experiments to show that their predictions are accurate. The authors reference the current published literature appropriately. The manuscript was well written, clear to understand, and thorough as to their motivation, previous literature, and techniques used. I additionally appreciated the inclusion of the mined database as well as the effort to make this reachable/useable for the community through a webserver. This is really an excellent job of streamlining or applying structural biology to a problem.

Reply: We thank the reviewer for the very kind words, appreciation and for providing suggestions for improvement.

My largest concern in this work is that there's already a confirmation bias at work in this parameterization, since the only information to draw from are reported labels that have been published and, therefore, have "worked." To test the robustness of the methodology, it would be useful to have a negative control or to include data from tags which have not worked in a way where it can be used to strengthen predictions (ex., how protein force fields are parameterized to capture minima AND barriers).

Reply: We agree with the referee that the included literature was biased toward residues that could be labeled. To address the referee's concern, we performed additional experiments, where we obtained and later characterized ten new single-cysteine variants of the maltose binding protein MalE during this revision. The new residues were randomly selected from residues within the best and worst 10% of all LS scores for MalE and we performed experiments to thoroughly characterize all new mutations in terms of protein expression level and degree of labelling.

Additionally, I think the authors should expand the discussion to report on the quality of the information conveyed by a specific label, and not just the ability of the label to be covalently attached. Should part of this guidance speak to the success of the information you get out of the label (is it fit for purpose?), instead of only the success of labeling the protein? I guess the argument could be that this provides a robust mechanism for making suggestions, and the researcher needs to make the final decision based on what they are interested in measuring, but the ability to do the latter negates the need for the former. This is touched on as point (iii) on page 10, but should be expanded on in the discussion of FRET benchmarking and algorithm development.

Reply: We emphasize that our algorithm does not consider fluorophore properties for the calculation of the label score (at the moment). For FRET and the computed "FRET score" our method does take into account specific Förster radii and performs coarse-grained modelling to obtain values for interdye distances. To address the reviewer's comment, we extended our discussion as follows: "We show using data from the literature and complementary experiments the predictive power of this labeling score and extend the method to systematically select residue pairs for FRET experiments, which we believe can be extended at later stage to consider the specific properties of the label and also other biophysical assays beyond FRET."

Minor points:

Figure 1A should be boxed in to separate and emphasize the parts of the Labelizer procedure in B and C. The A panel is a summary of problems labeling may introduce into assays – perhaps boxing it in and labeling it as such would make the figure clearer. Fluorescence property should be properties. Graphs in Panel C, parts 2 and 3 should have labeled axes (I am assuming 2 is residue # and I do not know what 3 is – probability distribution along residues?)

Reply: We have adapted Figure 1.

Supplementary Figure 1A should read "Labeled Residues per protein" on the X-axis. Soluability should be solubility.

Reply: We have corrected the mistakes.

Supplementary Figure S8 should have Angstroms instead of A.

Reply: We use the correct symbol "Å" for "Ångström" / Angstrom now in Supplementary Figure S8.

Reviewer #3 (Remarks to the Author):

This manuscript "Labelizer: systematic selection of protein residues for covalent fluorophore labeling" by Gebhardt, C. et al has described a general and, as claimed, quantitative strategy to identify optimal residues for protein labeling using an approach called naïve Bayes classifier. As described, the manuscript analyzed available literature and bioinformatics of more than 100 proteins with about 400 successfully labeled residues, from which the authors claimed that this strategy can be used to determine the labeling score to rank residues for their suitability as a labeling site. The approach was then expanded to a pair of residues for the systematic selection of a residue pair for FRET experiments. Finally, a Python package, which they call "labelizer", is developed to make the method available to a larger research community with anticipation of building a central open-access database for both successfully and non-successfully labeled proteins/residues. An algorithm to determine the fluorophore(s) labeling site with little to no effect on protein folding and function is definitely a very desirable tool in the fluorescence/FRET community. Figuring out the labeling sites at a distance of Förster radius in a protein is another sensitive tool for studying protein-ligand and other biomolecular interactions that encompasses small to large conformational changes of proteins. Overall, I find this manuscript interesting and well-written and I believe that it will be a very helpful resource for those who are seeking protein labeling. However, I also noted some issues (as outlined below) that need to be addressed.

Reply: We thank the reviewer for carefully reading the manuscript, the overall positive assessment, and for providing suggestions for improvement.

1) Provide a reference for 'Förster Radius' in this part of the sentence ".....but that are also compatible with the assay requirements, e.g., for FRET to result in an inter-fluorophore distance close to the Förster Radius R_0 ."

Reply: Following the referee's suggestion, we have added references for the Förster radius.

2) What does the -ve value of the conservation score in Fig 2A represent?

Reply: Figure 2A shows the conservation score as computed by the ConSurf server. Negative values correspond to highly conserved residues (often, the values are transformed in a linear way into conservation scores ranging from 1 (not conserved) to 9 (highly conserved)). We now explicitly state this in the caption Figure 2.

3) Based on the discussion related to Fig 2, it is my understanding that the low correlation parameters are preferred among different parameters (CS, SE, CR, and SS) to achieve successful labeling. However, the manuscript does not clarify why this is the case.

Reply: The referee points out correctly that parameter combinations are preferred that show little to no correlation with others since this is a prerequisite for Bayes analysis. We extended a statement related to this: "For this we calculated the correlation between all parameters to judge their statistical independence, which is desirable for our analysis (Figure 2B)."

4) The parameters ' \cap ' and ' Π ' in equation 3 are not defined.

Reply: The Greek symbol Π represents a product. In probability theory the symbol used to indicate an intersection is \cap , which refers to an element which is part of two first distinct sets. We have clarified this in the revised version of the main text.

5) To help readers better understand the manuscript, the authors should discuss the physical meaning of the critical parameters. For example, what does the label score value of 1 mean in Figure 3? Similarly, what does the FRET Score (FS) value of 1

mean in Fig 6? This applies to other parameters as well. This information will be a reference point for readers to understand the data.

Reply: While the selected four parameters in Figure 1 have a physical meaning, the numbers in Figure 3 or 5 cannot be directly used to say e.g., a value of 1 means solvent-exposed. For FRET-scores the value range additionally depends on the underlying LS values and thus no general recommendations are possible. The user of the algorithm has to inspect the distributions. In the text we added an entire section in the discussion to address this comment starting with the following sentences: "A challenging aspect of our analysis is the final selection of residues by the user based on the labelizer output. Since this step is decisive for which residues are used in experiments, the selection goes hand in hand with an assessment and interpretation of the LS/FS value distributions of the analyzed protein....."

6) What does "Label (x100)" represent in Fig. 3C?

Reply: The data set was multiplied with a factor of 100 to allow for better comparison of the two data sets, since they have very different overall residue numbers. We have inserted an explanation into the figure caption in the revised version.

7) Panel E, Fig 6, y-axis label: a space seems to be missing between "Deviation" and "of".

Reply: We have corrected the mistake.

8) On Page 10, in the paragraph "Our analysis shows, however, no correlation between LS and the experimentally determined DOL (Figure 4C). This is likely since all residues tested have relatively high label scores and we focused on mutants with a reasonable chance of labeling and did not include measurements e.g., of buried residues with low label scores." I believe that an actual correlation should be observed to claim that the prediction tool actually works. The explanation provided is not satisfactory to support the claim. At least, I found that the claim is not easy to understand.

Reply: As a result of this comment and those of referees #1 and #2, we have obtained and added to the manuscript new data for ten additional Male variants with randomly chosen labeling sites. In our new random data set we found that exclusively 'negative' variants with low LS values < 0.3 showed failure in fluorophore labelling or protein expression.

9) Data in Fig. 6F seems incomplete as the first concentration of phosphate used already shows the r_{closed} value of ~ 0.5 . Can the authors try a couple of lower phosphate concentrations and update the graph? Also, is the r_{closed} value determined to be zero via analysis (or experiment) in the absence of phosphate or this is an assumption? This has to be clarified.

Reply: We have conducted new experiments and updated the figure with a new titration of the same variant where more concentration values with r_{closed} around 0.5 were added. The experiments were done in technical triplicates.

10) The manuscript does not explain some of the important observations and I suggest that they should not be left uninterpreted. For example, Fig. 4 panel D, why the Cy3 labeling score is significantly lower as compared to all other fluorophores.

Reply: We observed experimentally that the degree of labelling (DOL) was lower for sCy3 in comparison to other dyes in the comparison provided with the original submission. In our lab, we have recently ordered a new sCy3-Maleimide batch for protein labelling and we do not observe this difference to other dyes anymore. This effect might thus be due to lower reactivity of the previous batch of dye used, yet we cannot make any repeats to support this claim (since there is no dye of the previous batch left anymore) and thus refrain from further interpretations.

Similarly, in Fig. 3, panel D, the sudden downward trend (red fit after around 100 Å) seems rather unusual given the upward trend of the raw data. This needs to be explained in the figure legend.

Reply: We assume that this comment referred to Figure 5D. The data >100 Å are noisy due to low statistics. As requested we now explicitly mention this and provide the reason in the caption of Figure 5.

11) Understandably the developed tool "labelizer" utilizes careful use of parameters and multiple intertwined analyses with boundaries and so on. This is something hard to fully assess by reviewers. The authors should make every effort to ensure that there are no errors in the tool and that what it predicts is fully reliable.

Reply: We are convinced that the updated version of our manuscript provides further guidance for users and adds new data for confidence on the use of the algorithm.